# Prevalence of Recurrent Mutations Predisposing to Breast Cancer in Early-Onset Breast Cancer Patients from Poland

**DOI:** 10.3390/cancers12082321

**Published:** 2020-08-17

**Authors:** Emilia Rogoża-Janiszewska, Karolina Malińska, Cezary Cybulski, Anna Jakubowska, Jacek Gronwald, Tomasz Huzarski, Marcin Lener, Bohdan Górski, Wojciech Kluźniak, Helena Rudnicka, Mohammad R. Akbari, Aniruddh Kashyap, Steven A. Narod, Jan Lubiński, Tadeusz Dębniak

**Affiliations:** 1Department of Genetics and Pathology, International Hereditary Cancer Center, Pomeranian Medical University, 71-252 Szczecin, Poland; karolina.malinska@pum.edu.pl (K.M.); cezarycy@pum.edu.pl (C.C.); aniaj@pum.edu.pl (A.J.); jgron@pum.edu.pl (J.G.); huzarski@pum.edu.pl (T.H.); marcinlener@poczta.onet.pl (M.L.); gorskib@pum.edu.pl (B.G.); kluzniak.w@gmail.com (W.K.); helena.rudnicka@pum.edu.pl (H.R.); kashyap@pum.edu.pl (A.K.); lubinski@pum.edu.pl (J.L.); debniak@pum.edu.pl (T.D.); 2Women’s College Research Institute, Women’s College Hospital, University of Toronto, Toronto, ON M5G 1N8, Canada; mohammad.akbari@utoronto.ca (M.R.A.); Steven.Narod@wchospital.ca (S.A.N.); 3Dalla Lana School of Public Health, University of Toronto, Toronto, ON M5T 3M7, Canada

**Keywords:** BRCA1, BRCA2, PALB2, CHEK2, NBN, RECQL, breast cancer, hereditary

## Abstract

There are twenty recurrent mutations in six breast-cancer-predisposing genes in Poland (BRCA1, BRCA2, CHEK2, PALB2, NBN, and RECQL). The frequencies of the twenty alleles have not been measured in a large series of early-onset breast cancer patients from Poland unselected for family history. We genotyped 2464 women with breast cancer diagnosed below age 41 years for twenty recurrent germline mutations in six genes, including BRCA1, BRCA2 CHEK2, PALB2, NBN, and RECQL. A mutation in one of the six genes was identified in 419 of the 2464 early-onset breast cancer cases (17%), including 22.4% of those cases diagnosed below age 31. The mutation frequency was 18.8% for familial breast cancer cases and 6% for non-familial cases. Among women with breast cancer below age 31, the mutation frequency was 23.6% for familial cases and 17.4% in non-familial cases. The majority of mutations (76.2%) were seen in BRCA1 and BRCA2. In Poland, a panel of twenty recurrent mutations in six genes can identify a genetic basis for a high percentage of early-onset cases and testing is recommended for all women with breast cancer at age 40 or below.

## 1. Introduction

Breast cancer is diagnosed in 2.1 million women each year and results in 600,000 deaths [1]. In Poland, the incidence of breast cancer is 93.42 per 100,000 per year and is relatively low by European standards [2]. Breast cancer accounts for about one-fourth of all cancer cases diagnosed in Polish women, and approximately one thousand women under the age of 39 were diagnosed with breast cancer in Poland in 2016 [2]. The incidence of breast cancer among premenopausal women has increased by approximately 1.5-fold over the past three decades; however, the incidence of breast cancer in young women diagnosed at age less than 40 years is relatively stable around the world [3].

There are two very important breast cancer susceptibility genes, BRCA1 and BRCA2. Other genes which were reported to be associated with breast cancer susceptibility include CHEK2, PALB2, TP53, PTEN, CDH1, NBN, ATM RAD50, BARD1, and RECQL [3,4,5].

Poland is homogeneous from a genetic perspective and the range of mutant alleles is limited, and a high percentage of all mutations are founder mutations. In ethnically-homogeneous populations, it is valuable to identify such common founder mutations in cancer-predisposing genes which may facilitate genetic testing. In our previous study, twenty recurrent mutations were found in six breast-cancer-predisposing genes (BRCA1, BRCA2, CHEK2, PALB2, NBN, and RECQL) in Polish breast cancer patients [4]. The twenty founder mutations were detected in 421 of 1018 (41%) Polish women with a strong family history of breast cancer (the mean number of breast cancers per family was 3.6 and the mean age of diagnosis of the probands was 44 years), including BRCA1/2 alleles seen in 354 cases and non-BRCA1/2 mutations seen in 67 cases [4]. The thirteen BRCA1 and BRCA2 mutations in the panel represented 84% of all BRCA1/2 mutations and the three CHEK2 truncating mutations constituted 95% of all CHEK2 mutations. The two recurrent PALB2 mutations represented 83% of all PALB2 mutations [4]. In total, the panel covered about 80% of all BRCA1/2, CHEK2, PALB2, NBN, and RECQL mutation detection in Polish high-risk families with breast cancer.

However, the frequencies of these twenty mutant alleles of BRCA1/2, CHEK2, PALB2, NBN, and RECQL have not been measured in a large series of early-onset breast cancer patients from Poland.

The goal of the current study was to estimate the prevalence of these twenty alleles in six genes in Polish early-onset breast cancer patients.

## 2. Results

A positive family history of cancer was observed in 1661 of 2354 (70.6%) patients diagnosed at age 40 or below, including 271 out of 340 (79.7%) patients diagnosed at age 30 or below (Table 1 and Table 2). Family history was missing for 110 cases.

We found a pathogenic mutation in 419 of the 2464 early-onset breast cancer cases (17%) (Table 3). The mutation frequency was 22.4% for those diagnosed at age 30 or below (Table 4). Among 2464 cases diagnosed at age 40 or below, 18.8% of familial cases and 6% of non-familial cases carried a mutation (Table 1). Among 340 breast cancer cases diagnosed at age 30 or below, 23.6% of familial cases and 17.4% of non-familial cases carried a mutation (Table 2).

As expected, the majority of mutations (72.3%) were in BRCA1 and BRCA2. We identified a BRCA1 mutation in 301 out of 2464 (12.2%) cases and a BRCA2 mutation in two of the cases (0.08%).

Seven recurrent mutations of CHEK2, PALB2, NBN, and RECQL represented the other 27.7% of all mutation-positive cases. Three different truncating mutations in CHEK2 were seen in 77 cases (3.1%). A PALB2 mutation was identified in 19 women (0.8%), and a RECQL recurrent variant was detected in two women (0.08%) (Table 3).

We calculated the odds ratios for early breast cancer for selected founder mutations by comparing mutation frequencies observed in this study in patients diagnosed at age 40 or below to those reported in a large series of population controls in our previous studies [3,5,6,7,8,9,10,11,12]. The odds ratios for breast cancer diagnosed at age 40 or below were 28.8 for BRCA1 (95% CI 18.6–44.5); 3.8 for CHEK2 (95% CI 2.5–5.5); and 3.6 for PALB2 (95% CI 1.6–7.5), and all were statistically significant. The odds ratio for NBN was 1.3 (95% CI 0.7–2.5), and it was not significant (*p* = 0.5). The odds ratio for RECQL was 1.9 (95% CI 0.3–13.6), and it was also not significant (*p* = 0.9) (Table 3).

## 3. Discussion

We found that one of the twenty mutations was present in 17% of breast cancer cases diagnosed at age 40 or below and in 22.4% of all breast cancer cases diagnosed at age 30 or below. BRCA1 mutations were more common (12.2% among breast cancer cases diagnosed at age 40 or below), while mutations in the BRCA2 gene were rare (0.08% in breast cancer patients diagnosed at age 40 or below). We confirmed a high risk of breast cancer among young BRCA1 female carriers (almost 30-fold increase for breast cancer at age 40 or below). This means that among women with a BRCA1 mutation in Poland, the risk of cancer at age 40 is much higher than that in the general population, but the odds ratio cannot be generalized to older women.

Truncating mutations of the CHEK2 gene (c.444+1G>A, c.1100delC, del5395(exon10-11del)), cause a two- to three-fold risk of breast cancer [5,13]. Among Polish women who carry a truncating CHEK2 mutation without a family history of breast cancer, the risk increase is about three-fold, and the risk is higher (increased about five-fold) in carriers of a truncating mutation of CHEK2 in the presence of positive family history of breast cancer [4]. In our study we found that the risk increase, given a CHEK2 truncating mutation, begins below age 40 (OR = 3.8; 95% CI 2.5–5.6).

Mutations in the NBN gene (also called NBS1) are responsible for the Nijmegen breakage syndrome (NBS), which is characterized by spontaneous chromosomal instability, immunodeficiency, and a predisposition to cancer. Approximately 90.0% of NBS patients carry the homozygous mutation 657del5 (c.657_661delACAAA) in exon 6, which has been predominantly identified in Slavic populations and confers a two-fold risk for breast cancer development [14]. In a meta-analysis of seven studies, strong evidence was presented for a moderately increased breast cancer risk associated with a truncating variant, c.657del5, with a relative risk of 2.7 [15]. Another study from Poland provided evidence for an association between the variant and early-onset breast cancer [16]. In our study we found that the risk of breast cancer <41 years was not increased (OR = 1.3; 95% CI 0.7–2.5).

Approximately one percent of cases had one of two recurrent mutations in the PALB2 gene. In Finland, Canada, and Poland, these two mutations are present in 0.5–1.0% of all breast cancers [17,18,19]. PALB2 mutations have been associated with a relative risk up to five-fold, although studies in different populations have produced divergent estimates [6,18,19,20]. In Poland, two founder PALB2 mutations were detected (c.509_510delGA, c.172_175delTTGT) and were associated with a 4.5-fold increased risk of breast cancer in unselected Polish women [4,21] similar to the odds ratio we found for young breast cancer patients (OR = 3.6; 95% CI 1.7–7.9).

In this study we found that the risk increase of breast cancer development in carriers of the RECQL mutation is 1.9 (95% CI 0.27–13.6), but the mutation was rare (two cases) and the odds ratio was not statistically significant. The RECQL gene mutation was previously reported to increase the risk of breast cancer by five-fold among unselected cases from Poland [4].

It seems reasonable to use the panel of twenty Polish founder mutations as a test for all young breast cancer patients. Our panel has a high sensitivity. We propose to use this panel as an inexpensive prescreening tool. Full sequencing (in particular, using gene panels including BRCA1/2) is warranted in women with early-onset breast cancer, in particular for those with a positive family history and those with breast cancers with clinical features characteristic of BRCA1-related tumors. The decision of management with mutation carriers will depend on the type of the detected mutation and on the clinical and pedigree data, i.e., in case of a BRCA1/2 mutation, appropriate surveillance and preventive surgery are advised. The identification of a BRCA1/2 mutation at the time of diagnosis can also have an impact on selecting patients for targeted therapy. Cisplatin has been reported to be effective and generally well tolerated among Polish BRCA1 patients with breast cancer [22,23]. Mutations in CHEK2 have been associated with resistance to anthracycline-based chemotherapy in breast cancer patients [24], but further studies are needed.

Of note, we observed that the frequencies of the twenty predisposing mutations were higher (43%) in families with strong breast cancer clustering than in women with early-onset breast cancer (17%) [4] (Table 5). Therefore, strong family history seems to be a stronger indicator of heredity than early age of diagnosis.

## 4. Materials and Methods

### 4.1. Cases

The women with breast cancer were selected from a registry of 25,000 breast cancer cases housed at the Hereditary Cancer Center in Szczecin. Patients were diagnosed in 16 different centers in Poland between 2000 and 2017. All cases were confirmed to have invasive breast cancer through review of the pathology reports.

Patients were invited to participate and provide blood samples within 12 months from diagnosis. During the patient interview, the goals of the study were explained, informed consent was obtained, genetic counseling was given, and a blood sample taken for DNA analysis. All patients and control subjects were of European ancestry and ethnic Poles. Family histories were collected during an interview. A positive family history was defined as the occurrence of one or more breast cancers among first or second degree relatives. For this study we included 2464 consecutive patients diagnosed with early-onset breast cancer at age 40 or below (mean age 37, range 18–40).

The study conformed to the Declaration of Helsinki, and all participants signed an informed consent document prior to participation. The study was approved by the institutional review board of the Pomeranian Medical University (KB-0012/97/17, approval date 19 June 2017).

### 4.2. Analysis of the Most Common BRCA Founder Mutations

DNA was isolated from blood taken from the participants using standard methods. Peripheral blood leukocytes were isolated and subsequent DNA extraction undertaken in the Department of Genetics and Pathology in Szczecin.

Genetic tests for three common Polish BRCA1 mutations were performed using multiplex-polymerase chain reaction assay. The 5382insC variant in exon 20 and the 4153delA variant in exon 11 were detected using an allele-specific amplification PCR (ASA-PCR). The last recurrent mutation c.181T>G in exon 5 of BRCA1 was detected using restriction fragment length polymorphism PCR (RFLP-PCR) and Ava II enzyme [3]. All other recurrent mutations in the BRCA1 gene (c.66_67delAG, c.3700_3704delGTAAA, c.5251C>T, c.676delT, c.1687C>T, c.3756_3759delGTCT) and BRCA2 gene (c.3847_3848delGT, c.7910_7914delCCTTT, c.658_659delGT, c.5946delT) were genotyped using TaqMan assay (Applied Biosystems/Life Technologies, Foster City, CA, USA) the LightCycler Real-Time PCR 480 system (Roche Life Science, Penzberg, Germany). The primer and probe sequences are available upon request. Laboratory technicians were blinded to case-control status.

### 4.3. Analysis of the Non-BRCA Founder Mutations

Other founder genes, CHEK2 (c.1100delC, c.444+1G>A, del5395), PALB2 (c.509_510delGA, c.172_175delTTGT), NBN 657del5 (c.657_661delACAAA), and RECQL (c.1667_1667+3delAGTA), were genotyped using TaqMan assay (Applied Biosystems/Life Technologies) and the LightCycler Real-Time PCR 480 system (Roche Life Science). The primer and probe sequences are available upon request. Laboratory technicians were blinded to case-control status. A large deletion of exon 9–10 of CHEK2 gene was genotyped using multiplex-PCR reaction. This multiplex-PCR method was based on our report in 2000 and validated later [4,7,8,9].

In order to confirm germline recurrent small mutations in the BRCA1, BRCA2, CHEK2, PALB2, NBN, and RECQL genes, Sanger sequencing was performed from a second independent blood sample. The sequencing reaction was performed using a BigDye Terminator v3.1 Cycle Sequencing Kit (Life Technologies). Sequencing products were analyzed using a genetic analyzer ABI Prism 3500XL (Life Technologies). All BRCA1/2, CHEK2, PALB2, NBN, and RECQL sequences were compared to the NCBI reference sequence (RefSeq) reported in GenBank (https://www.ncbi.nlm.nih.gov/genbank/): NM_007294.3, NM_000059.3, NM_001005735.1, NM_024675.3, NM_002485.4, NM_002907.3.

### 4.4. Statistical Analysis

We estimated the odds ratios for selected recurrent mutations to investigate their association with early-onset breast cancer risk. To do so, we compared mutation frequencies in patients with breast cancer (cases) to those seen in Polish cancer-free individuals (controls). For BRCA1, CHEK2, NBN, PALB2, BLM, and RECQL, as a reference we used mutation frequencies in a large series of controls from our previous studies [4,8,9,10,11,12,13,21,25]. Odds ratios (OR) were generated from two-by-two tables, and statistical significance was assessed with the Fisher exact test or the Chi-squared test where appropriate. The odds ratios were generated for women with breast cancer below age 41 and for women with breast cancer below age 31.

## 5. Conclusions

The goal of the current study was to estimate the prevalence of twenty alleles in six genes, BRCA1/2, CHEK2, PALB2, NBN, and RECQL, in Polish early-onset breast cancer patients. In the current study we found that this panel of twenty recurrent mutations can identify a genetic basis for approximately one-fifth of early-onset cases, and testing is recommended for all women with breast cancer under age 40. The identification of a mutation at the time of diagnosis can have an impact on selecting patients for preventive surgery or targeted therapy.

## Figures and Tables

**Table 1 cancers-12-02321-t001:** The frequencies of recurrent mutations in Polish women with breast cancer diagnosed at age <41 years, by family history of breast cancer.

Gene	Variant	Family History	
		Positive (*n* = 1661)	Negative (*n* = 693)
BRCA1	c.5266dupC	141 (8.5%)	11 (1.6%)
	c.181T>G	65 (3.9%)	5 (0.7%)
	c.4035delA	8 (0.5%)	0 (0%)
	c.66-67delAG	6 (0.4%)	0 (0%)
	c.3700_3704delGTAAA	8 (0.5%)	1 (0.1%)
	c.5251C>T	5 (0.3%)	0 (0%)
	c.5346G>A	1 (0.06%)	0 (0%)
		All BRCA1	301 (12.2%)
BRCA2	c.3847_3848delGT	1 (0.06%)	0 (0%)
	c.658_659delGT	1 (0.06%)	0 (0%)
		All BRCA2	2 (0.08%)
CHEK2	c.1100delC	11 (0.7%)	5 (0.7%)
	c.444+1G>A	28 (1.7%)	6 (0.9%)
	del5395	16 (1%)	3 (0.4%)
		All CHEK2	77 (3.1%)
PALB2	c.509_510delGA	5 (0.3%)	1 (0.1%)
	c.172_175delTTGT	8 (0.5%)	5 (0.7%)
		All PALB2	19 (0.8%)
NBN	c.657_661delACAAA	8 (0.5%)	5 (0.7%)
		All NBN	18 (0.7%)
RECQL	c.1667_1667+3delAGTA	1 (0.06%)	1 (0.1%)
		All RECQL	2 (0.08%)
Total number of carriers		313 (18.8%)	43 (6.2%)

Family cancer history was missing for 110 cases diagnosed at age beetwen 31 and 40.

**Table 2 cancers-12-02321-t002:** The frequencies of recurrent mutations in Polish women with breast cancer diagnosed at age <31 years, by family history of breast cancer.

Gene	Variant	Family History	
		Positive (*n* = 271)	Negative (*n* = 69)
BRCA1	c.5266dupC	28 (10.3%)	7 (10.2%)
	c.181T>G	12 (4.4%)	1 (1.4%)
	c.4035delA	2 (0.7%)	0
	c.66-67delAG	1 (0.3%)	0
	c.3700_3704delGTAAA	2 (0.7%)	0
	c.5251C>T	1 (0.4%)	0
	c.5346G>A	1 (0.4%)	0
		All BRCA1	55(16%)
BRCA2	c.658_659delGT	1 (0.4%)	0
		All BRCA2	0
CHEK2	c.1100delC	0	1 (1.4%)
	c.444+1G>A	9 (3.3%)	1 (1.4%)
	del5395	2 (0.7%)	0
		All CHEK2	13 (3.8%)
PALB2	c.509_510delGA	1 (0.4%)	0
	c.172_175delTTGT	3 (1.1%)	2 (2.9%)
		All PALB2	6 (1.8%)
RECQL	c.1667_1667+3delAGTA	1 (0.4%)	0
		All RECQL	1 (0.3%)
Total number of carriers		64 (23.6%)	12 (17.4%)

Family cancer history was available for all cases diagnosed at age 30 or below.

**Table 3 cancers-12-02321-t003:** The frequencies of recurrent mutations in Polish women with breast cancer diagnosed at age <41 years and in controls, with corresponding odds ratios.

Gene	Variant	Classification	Total Carriers (*n* = 2464)	Controls	Odds Ratio (95% CI)	*p*-Value
BRCA1	c.5266dupC	frameshift_variant	184 (7.5%)	17/4570 (0.4%)	21.6 (13.116–35.617)	<0.0001
	c.181T>G	missense	83 (3.4%)	3/4570 (0.07%)	53.1 (16.749–168.14)	<0.0001
	c.4035delA	frameshift_variant	13 (0.5%)	2/4570 (0.04%)	12.1 (2.731–53.745)	<0.0001
	c.66-67delAG	frameshift_variant	6 (0.2%)	-	-	-
	c.3700_3704delGTAAA	frameshift_variant	9 (0.3%)	-	-	-
	c.5251C>T	stop_gained	5 (0.2%)	-	-	-
	c.5346G>A	stop_gained	1 (0.04%)	-	-	-
		All BRCA1	301 (12.2%)	22/4570 (0.5%)	28.8 (18.602–44.489)	<0.0001
BRCA2	c.3847_3848delGT	frameshift_variant	1 (0.04%)	-	-	-
	c.658_659delGT	frameshift_variant	1 (0.04%)	-	-	-
		All BRCA2	2 (0.08%)	-	-	-
CHEK2	c.1100delC	frameshift_variant	16 (0.6%)	7/4346 (0.2%)	4.1 (1.664–9.863)	0.0018
	c.444+1G>A	splice_donor_variant	39 (1.6%)	14/4346 (0.3%)	5.0 (2.696–9.185)	<0.0001
	del5395	frameshift_variant	22 (0.9%)	16/4346 (0.4%)	2.4 (1.278–4.652)	0.0087
		All CHEK2	77 (3.1%)	37/4346 (0.9%)	3.8 (2.530–5.578)	<0.0001
PALB2	c.509_510delGA	frameshift_variant	12 (0.5%)	7/4702 (0.1%)	3.3 (1.290–8.350)	0.0163
	c.172_175delTTGT	frameshift_variant	7 (0.3%)	3/4702 (0.1%)	4.5 (1.153–17.278)	0.0414
		All PALB2	19 (0.8%)	10/4702 (0.2%)	3.6 (1.692–7.855)	0.0008
NBN	c.657_661delACAAA	frameshift_variant	18 (0.7%)	22/4000 (0.6%)	1.3 (0.7122–2.486)	0.4620
		All NBN	18 (0.7%)	22/4000 (0.6%)	1.3 (0.7122–2.486)	0.4620
RECQL	c.1667_1667+3delAGTA	splice_donor_variant	2 (0.08%)	2/4702 (0.04%)	1.9 (0.2686–13.567)	0.8956
		All RECQL	2 (0.08%)	2/4702 (0.04%)	1.9 (0.2686–13.567)	0.8956
Total number of carriers			419 (17%)			

**Table 4 cancers-12-02321-t004:** The frequencies of recurrent mutations in Polish women with breast cancer diagnosed at age <31 years and in controls, with corresponding odds ratios.

Gene	Variant	Classification	Total Carriers (*n* = 340)	Controls	Odds Ratio (95% CI)	*p*-Value
BRCA1	c.5266dupC	frameshift_variant	35 (10.3%)	17/4570 (0.4%)	30.7 (17.019–55.502)	<0.0001
	c.181T>G	missense	13 (3.8%)	3/4570 (0.07%)	60.5 (17.155–213.51)	<0.0001
	c.4035delA	frameshift_variant	2 (0.6%)	2/4570 (0.04%)	13.5 (1.897–96.288)	0.0160
	c.66-67delAG	frameshift_variant	1 (0.3%)	-	-	-
	c.3700_3704delGTAAA	frameshift_variant	2 (0.6%)	-	-	-
	c.5251C>T	stop_gained	1 (0.3%)	-	-	-
	c.5346G>A	stop_gained	1 (0.3%)	-	-	-
		All BRCA1	55 (16%)	22/4570 (0.5%)	39.9 (23.985–66.359)	<0.0001
BRCA2	c.658_659delGT	frameshift_variant	1 (0.3%)	-	-	-
		All BRCA2	1 (0.3%)	-	-	-
CHEK2	c.1100delC	frameshift_variant	1 (0.3%)	7/4346 (0.2%)	1.8 (0.2242–14.913)	0.5671
	c.444+1G>A	splice_donor_variant	10 (2.9%)	14/4346 (0.3%)	9.4 (4.132–21.277)	<0.0001
	del5395	frameshift_variant	2 (0.6%)	16/4346 (0.4%)	1.6 (0.3665–6.996)	0.8598
		All CHEK2	13 (3.8%)	37/4346 (0.9%)	4.6 (2.436–8.798)	<0.0001
PALB2	c.509_510delGA	frameshift_variant	1 (0.3%)	7/4702 (0.1%)	1.98 (0.2426–16.136)	0.5158
	c.172_175delTTGT	frameshift_variant	5 (1.5%)	3/4702 (0.1%)	23.4 (5.561–98.278)	<0.0001
		All PALB2	6 (1.8%)	10/4702 (0.2%)	8.4 (3.044–23.338)	<0.0001
NBN	c.657_661delACAAA	frameshift_variant	0	22/4000 (0.6%)	-	-
		All NBN	0	22/4000 (0.6%)	-	-
RECQL	c.1667_1667+3delAGTA	splice_donor_variant	1 (0.3%)	2/4702 (0.04%)	6.9 (0.6266–76.688)	0.4930
		All RECQL	1 (0.3%)	2/4702 (0.04%)	6.9 (0.6266–76.688)	0.4930
Total number of carriers			76 (22.4%)			

**Table 5 cancers-12-02321-t005:** The frequencies of recurrent mutations in Polish women with early-onset breast cancer and in Polish women with hereditary breast cancer (HBC).

Gene	Variant	Classification	Carriers <41 years (*n* = 2464)	HBC Carriers *
BRCA1	c.5266dupC	frameshift_variant	184 (7.5%)	204/1018 (20%)
	c.181T>G	missense	83 (3.4%)	84/1018 (8.3%)
	c.4035delA	frameshift_variant	13 (0.5%)	15/1018 (1.5%)
	c.66-67delAG	frameshift_variant	6 (0.2%)	-
	c.3700_3704delGTAAA	frameshift_variant	9 (0.3%)	10/1018 (1%)
	c.5251C>T	stop_gained	5 (0.2%)	6/1018 (0.6%)
	c.5346G>A	stop_gained	1 (0.04%)	5/1018 (0.5%)
		All BRCA1	301 (12.2%)	324 (31.8%)
BRCA2	c.3847_3848delGT	frameshift_variant	1 (0.04%)	-
	c.658_659delGT	frameshift_variant	1 (0.04%)	3/1018 (0.3%)
		All BRCA2	2 (0.08%)	3/1018 (0.3%)
CHEK2	c.1100delC	frameshift_variant	16 (0.6%)	8/598 (1.3%)
	c.444+1G>A	splice_donor_variant	39 (1.6%)	13/598 (2.2%)
	del5395	frameshift_variant	22 (0.9%)	14/598 (2.3%)
		All CHEK2	77 (3.1%)	35 (5.9%)
PALB2	c.509_510delGA	frameshift_variant	12 (0.5%)	12/598 (2%)
	c.172_175delTTGT	frameshift_variant	7 (0.3%)	7/598 (1.2%)
		All PALB2	19 (0.8%)	19 (3.2%)
NBN	c.657_661delACAAA	frameshift_variant	18 (0.7%)	9/598 (1.5%)
		All NBN	18 (0.7%)	9 (1.5%)
RECQL	c.1667_1667+3delAGTA	splice_donor_variant	2 (0.08%)	4/598 (0.7%)
		All RECQL	2 (0.08%)	4 (0.7%)
Total number of carriers			17%	43.3%

* Mutation prevalence of selected variants in HBC patients were from our previous Polish study [4].

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
