# Peer review of "Prevalence of Recurrent Mutations Predisposing to Breast Cancer in Early-Onset Breast Cancer Patients from Poland"

_cancers, 2020, doi:10.3390/cancers12082321_

Round 1

Reviewer 1 Report

In the present manuscript, the authors further build on results that were recently published and that they co-authored (their ref 5). According to that publication, a simple test for 20 recurrent mutations in six genes will facilitate genetic testing for breast cancer susceptibility in Poland. This panel of mutations was deduced from data obtained with patients with a strong family history of breast cancer. In the present manuscript, authors applied this same panel to a cohort of BC patients diagnosed at young age (<41 y). As (they) expected, they recovered the set of 20 mutations in a high fraction (17%) of the samples (while in 50.3% of the samples when patient selection was based on strong family history according to ref 5). Authors also report that recovery rates increased in patients diagnosed at youngest age (<31 years) and mention further increased recovery rates when patients report a family history of BC. As a general conclusion, authors claim that “It seems reasonable to use such a panel as test for all young breast cancer patients and more extensive testing to be offered to those women with no mutation and a strong family history”.

The data have been correctly generated and analyzed.

Comments:

Line26-29: in the ABSTRACT, separate mutation frequencies are provided whether or not patients have a family history of BC. However, those data are not provided in the summary tables as results.

Line 30: should “hereditary” not be replaced by “genetic” as the selection criteria for screening was age at diagnosis and not a hereditary pattern of transmission?

Line 43-47: the manner genes previously linked the BC predisposition are sub-grouped in 2 categories, those with “strong evidence of pathogenicity” and those with “weak evidence of association with increased BC risk” is rather simplistic and may lead to confusion. Mutations with a strong evidence for pathogenicity may generate a rather weakly (but confirmed) increased BC risk. Moreover, for young BC patients that do not belong to families with a strong history of BC, there is an increased chance that carriership of several mutations contributed to BC initiation (polygenic model).

Line 51-52: the 6 genes with recurrent mutations that were included in the panel are called “high-risk BC predisposing genes in this sentence. For BRCA1/2 no discussion, but for the others?! A few lines before, RECQL was considered as presenting relatively weak evidence for association with increased BC risk!

Line 52-53: mutant allele frequencies have not yet been reported in early onset BC patient, but are known for high risk familial BC cases from Poland (ref 5). Mention these frequencies (globally, not necessarily for each mutations separately) in the INTRODUCTION.

Line 61-62: according to this sentence, mutation frequencies in cases with (or without) family history of BC are indicated in Table 1. This is not the case!

Line 62: According to main text, 271 BC cases were diagnosed with BC before age 31. However, in Table 2, 340 cases are mentioned!

Line 62-63:  Although stated in the main text, the separate mutation frequencies obtained for BC cases diagnosed <31 year with (or without) family history of BC are not provided Table 2.

Line 64: 303/419 corresponds to 72.3% (not “72.6%” as mentioned) which also better fits with the “27.7%” mentioned in line 67.

Line 70-71: the meaning and relevance of the message in this sentence is unclear! The sentence reports the frequency of a common CHEK2 missense mutation in the young BC samples. However, this variant does not belong to the panel of mutations investigated in this study. Also, the mentioned result is not further discussed in the manuscript.

Line 80-82: …to estimate…., with…. . Message unclear

Line 117-121: this information should be presented in the Introduction. Make also clear whether the mentioned frequencies relate to the fraction of mutations recovered with the panel compared to the whole set of mutations previously detected with the standard mutation screen, or to the fraction of BC patients in which a mutation can be detected with the panel compared to the standard mutation screen. Both types of frequencies are useful and should be mentioned somewhere in the manuscript (or Table).

Line 122-123: I have problems with this recommendation as it would mean that BC patients diagnosed at young age but without strong family history of BC cannot benefit anymore from the “Standard” genetic screen, as only the panel will be applied. The panel screen should rather be seen as a pre-screen: when the outcome is negative, the “standard” screen should further be performed. However, when the outcome of the panel screen is positive, other issues for discussion will emerge. For instance, what will be done if a founder mutation in RECQL is detected with the panel? If no further analyses are performed (what will probably happen), I’m afraid a BRCA1/2 mutation will be missed in a non negligible fraction of patients. There is indeed no indication/guarantee for mutual exclusivity between pathogenic mutations in BRCA1/2 and RECQL (and many other candidate BC predisposing genes).  Combined carriership may even be a requirement for strong increased BC risk for many candidate BC predisposing genes (acting according to polygenic models).

Line 124-128: In this paragraph the authors correctly highlight the clinical utility of this cheap test with short turnaround time for patients that just got a diagnosis and treatment options have to be considered at short term.  But how shall carriers (and non carriers belonging to a “positive” family) be further counseled, especially when a non BRCA1/2 was detected with the panel? According to the authors, the panel is designed to identify “recurrent mutations in six high-risk BC predisposing genes” (see lines 50-52). This may lead to confusions and even errors at the genetic counseling. Authors should clarify in the DISCUSSION how the outcomes of the panel screens will be translated during the genetic counseling.

Line 133: Young BC patients diagnosed between 2000 and 2017 were traced and when alive requested to participate in the study. Using this approach, predisposing mutations leading to poor survival might be under-represented.   Are the relative recovery rates among young patients and patients belonging to high risk families comparable?

Line 138-139: This sentence defines the expression “positive family history” for young BC cases. To allow comparison of some data presented in this manuscript, the definition of “family with strong history of BC” that was used to identify and quantify cancer predisposing mutations in Poland (and later on to establish the panel) should also be provided somewhere in the manuscript.

Line 187-190: These 3 sentences should not be presented as “conclusions”. The contained information should move to the INTRODUCTION part. Clinical implications should rather be presented in CONCLUSION.

TABLES:

  • Data allowing comparison between young BC cases with and without family history of BC are not presented in both tables, although this is indicated in the text.
  • Mutation recovery rates in families with a strong history of BC should be presented for comparison (in existing tables or a separate one).
  • According to Table 1 and Table 2, the c.172_175del mutation in PALB2 is found almost exclusively in very young BC patients (5/340 BC patients diagnosed before 31y; 2/2124 BC patients diagnosed between 31y and 40y). Is this correct? If so, is the same trend observed in families selected for strong history of BC?   

Author Response

Line26-29: in the ABSTRACT, separate mutation frequencies are provided whether or not patients have a family history of BC. However, those data are not provided in the summary tables as results.

We have included mutation frequencies by family history in Table 3 and Table 4 (Line26-29).

Line 30: should “hereditary” not be replaced by “genetic” as the selection criteria for screening was age at diagnosis and not a hereditary pattern of transmission?

We have replaced “hereditary” to “genetic” (line 30).

Line 43-47: the manner genes previously linked the BC predisposition are sub-grouped in 2 categories, those with “strong evidence of pathogenicity” and those with “weak evidence of association with increased BC risk” is rather simplistic and may lead to confusion. Mutations with a strong evidence for pathogenicity may generate a rather weakly (but confirmed) increased BC risk. Moreover, for young BC patients that do not belong to families with a strong history of BC, there is an increased chance that carriership of several mutations contributed to BC initiation (polygenic model).

We have combined all non-BRCA1/2 genes to one category (as genes reported to be associated with a susceptibility to breast cancer) (line 44-46).

Line 51-52: the 6 genes with recurrent mutations that were included in the panel are called “high-risk BC predisposing genes in this sentence. For BRCA1/2 no discussion, but for the others?! A few lines before, RECQL was considered as presenting relatively weak evidence for association with increased BC risk!

We agree with the reviewer.  We have deleted the term "high-risk" (line 51).

Line 52-53: mutant allele frequencies have not yet been reported in early onset BC patient, but are known for high risk familial BC cases from Poland (ref 5). Mention these frequencies (globally, not necessarily for each mutations separately) in the INTRODUCTION.

We have added requested data to the Introduction (line 52-55).

Line 61-62: according to this sentence, mutation frequencies in cases with (or without) family history of BC are indicated in Table 1. This is not the case!

We have added Table 1 and 2 with mutation frequencies in cases with/without family history of breast cancer and corrected the sentences in text (line 67-68).

Line 62: According to main text, 271 BC cases were diagnosed with BC before age 31. However, in Table 2, 340 cases are mentioned!

We have corrected this mistake (line 72).

Line 62-63:  Although stated in the main text, the separate mutation frequencies obtained for BC cases diagnosed <31 year with (or without) family history of BC are not provided Table 2.

We have added Table 2 (line 73).

Line 64: 303/419 corresponds to 72.3% (not “72.6%” as mentioned) which also better fits with the “27.7%” mentioned in line 67.

We have corrected this mistake (line 74).

Line 70-71: the meaning and relevance of the message in this sentence is unclear! The sentence reports the frequency of a common CHEK2 missense mutation in the young BC samples. However, this variant does not belong to the panel of mutations investigated in this study. Also, the mentioned result is not further discussed in the manuscript.

We have removed this sentence (line 78-79).

Line 80-82: …to estimate…., with…. . Message unclear

We have corrected this sentence (line 89-90).

Line 117-121: this information should be presented in the Introduction. Make also clear whether the mentioned frequencies relate to the fraction of mutations recovered with the panel compared to the whole set of mutations previously detected with the standard mutation screen, or to the fraction of BC patients in which a mutation can be detected with the panel compared to the standard mutation screen. Both types of frequencies are useful and should be mentioned somewhere in the manuscript (or Table).

As suggested by the reviewer, we have moved this information to the Introduction (line 55-60).

Line 122-123: I have problems with this recommendation as it would mean that BC patients diagnosed at young age but without strong family history of BC cannot benefit anymore from the “Standard” genetic screen, as only the panel will be applied. The panel screen should rather be seen as a pre-screen: when the outcome is negative, the “standard” screen should further be performed. However, when the outcome of the panel screen is positive, other issues for discussion will emerge. For instance, what will be done if a founder mutation in RECQL is detected with the panel? If no further analyses are performed (what will probably happen), I’m afraid a BRCA1/2 mutation will be missed in a non negligible fraction of patients. There is indeed no indication/guarantee for mutual exclusivity between pathogenic mutations in BRCA1/2 and RECQL (and many other candidate BC predisposing genes).  Combined carriership may even be a requirement for strong increased BC risk for many candidate BC predisposing genes (acting according to polygenic models).

We have responded to this issue in the text of the paper (line 125-128).

Line 124-128: In this paragraph the authors correctly highlight the clinical utility of this cheap test with short turnaround time for patients that just got a diagnosis and treatment options have to be considered at short term.  But how shall carriers (and non carriers belonging to a “positive” family) be further counseled, especially when a non BRCA1/2 was detected with the panel? According to the authors, the panel is designed to identify “recurrent mutations in six high-risk BC predisposing genes” (see lines 50-52). This may lead to confusions and even errors at the genetic counseling. Authors should clarify in the DISCUSSION how the outcomes of the panel screens will be translated during the genetic counseling.

We agree with the reviewer. Our panel includes different genes. We clarified  in the discussion the outcomes of the panel screens, i.e. we have explained clinical management when a BRCA1/2 mutation is detected (line 130-133).

Line 133: Young BC patients diagnosed between 2000 and 2017 were traced and when alive requested to participate in the study. Using this approach, predisposing mutations leading to poor survival might be under-represented.   Are the relative recovery rates among young patients and patients belonging to high risk families comparable?

We added to “Material and Methods”, that patients were invited to participate in the study within 12 months from diagnosis (line 147-148).

Line 138-139: This sentence defines the expression “positive family history” for young BC cases. To allow comparison of some data presented in this manuscript, the definition of “family with strong history of BC” that was used to identify and quantify cancer predisposing mutations in Poland (and later on to establish the panel) should also be provided somewhere in the manuscript.

We have corrected definition of our familial cases (line 151-152). Additionally we have added Table 5 with mutation frequencies in early-onset carriers and HBC cases  (from our previous study) [ref. 4].

Line 187-190: These 3 sentences should not be presented as “conclusions”. The contained information should move to the INTRODUCTION part. Clinical implications should rather be presented in CONCLUSION.

We have corrected our conclusions ( line 197-202).

TABLES:

    Data allowing comparison between young BC cases with and without family history of BC are not presented in both tables, although this is indicated in the text.

We have shown mutation frequencies in early onset cases, by family history of breast cancer, in Tables 3 and 4.

    Mutation recovery rates in families with a strong history of BC should be presented for comparison (in existing tables or a separate one).

We have shown mutation frequencies in early onset cases and in HBC cases in Table 5. We have quoted the table 5 in the discussion.

    According to Table 1 and Table 2, the c.172_175del mutation in PALB2 is found almost exclusively in very young BC patients (5/340 BC patients diagnosed before 31y; 2/2124 BC patients diagnosed between 31y and 40y). Is this correct? If so, is the same trend observed in families selected for strong history of BC?

This is correct. We can’t answer of this question, because in our previous study only 8 probands with strong history of breast cancer had breast cancer below age of 31 yrs. Neither carried a PALB2 mutation.

Reviewer 2 Report

This a survey of 6 common genes (BRCA1, BRCA2, CHEK2, PalB2 NBN, and RECQL) to assess the mutational frequencies in cohort of nearly 2500 Polish women with breast under age 41. This taken from a registry of 25,000 women with breast cancer at the Hereditary Cancer Center, which represents 16 centers throughout Poland between 2000 and 2017. All patients were prospectively consented. This paper represents a snap shot of the mutational landscape young Polish, which the authors say has not been done before. The odds ratios (compared to large series of population controls) were significant for BRAC1, CHEK2 and Palb2. NBN and RECQL were not significant. BRAC2 was rare present in 2 (0.8%). I think this is valuable contribution and I would accept it.   

Author Response

This a survey of 6 common genes (BRCA1, BRCA2, CHEK2, PalB2 NBN, and RECQL) to assess the mutational frequencies in cohort of nearly 2500 Polish women with breast under age 41. This taken from a registry of 25,000 women with breast cancer at the Hereditary Cancer Center, which represents 16 centers throughout Poland between 2000 and 2017. All patients were prospectively consented. This paper represents a snap shot of the mutational landscape young Polish, which the authors say has not been done before. The odds ratios (compared to large series of population controls) were significant for BRAC1, CHEK2 and Palb2. NBN and RECQL were not significant. BRAC2 was rare present in 2 (0.8%). I think this is valuable contribution and I would accept it.

We thank the reviewer for his positive review.

Reviewer 3 Report

Interesting article about the prevalence of recurrent mutations predisposing to breast cancer in early-onset breast cancer patients from Poland.

I recommend to emphasized the conclusion more clearly for clinicians ("the identification of a mutation at the time of diagnosis can impact on selecting patients for preventive surgery or targeted therapy")

by writing this sentence in the conclusion chapter.

Author Response

Interesting article about the prevalence of recurrent mutations predisposing to breast cancer in early-onset breast cancer patients from Poland.

I recommend to emphasized the conclusion more clearly for clinicians ("the identification of a mutation at the time of diagnosis can impact on selecting patients for preventive surgery or targeted therapy")by writing this sentence in the conclusion chapter.

We have added this sentence to conclusion chapter (line 201-202).

Reviewer 4 Report

The manuscript “Prevalence of recurrent mutations predisposing to breast cancer in early-onset breast cancer patients from Poland” by Rogoza-Janiszewska at el. describes the analysis of mutations present in six breast-cancer-predisposing genes among Polish female breast cancer patients diagnosed below the age of 41.  The topic of the manuscript is important and should meet the interest of readers of Cancers.  Below I am addressing some minor corrections for authors to include in order to improve the quality of their work.

  1. Please provide missing references for three first sentences in the Introduction section.
  2. Line 85, please correct the typo “cance”.

Author Response

The manuscript “Prevalence of recurrent mutations predisposing to breast cancer in early-onset breast cancer patients from Poland” by Rogoza-Janiszewska at el. describes the analysis of mutations present in six breast-cancer-predisposing genes among Polish female breast cancer patients diagnosed below the age of 41.  The topic of the manuscript is important and should meet the interest of readers of Cancers.  Below I am addressing some minor corrections for authors to include in order to improve the quality of their work.

    Please provide missing references for three first sentences in the Introduction section.

We have added requested references (line 36-43).

    Line 85, please correct the typo “cance”.

We have corrected the typo (line 91).

Round 2

Reviewer 1 Report

no further comments